# Alleviating Label Switching with Optimal Transport

**Pierre Monteiller**
ENS Ulm
pierre.monteiller@ens.fr

**Sebastian Claici**
MIT CSAIL & MIT-IBM Watson AI Lab
sclaici@mit.edu

**Edward Chien**
MIT CSAIL & MIT-IBM Watson AI Lab
edchien@mit.edu

**Farzaneh Mirzazadeh**
IBM Research & MIT-IBM Watson AI Lab
farzaneh@ibm.com

**Justin Solomon**
MIT CSAIL & MIT-IBM Watson AI Lab
jsolomon@mit.edu

**Mikhail Yurochkin**
IBM Research & MIT-IBM Watson AI Lab
mikhail.yurochkin@ibm.com

## Abstract

*Label switching* is a phenomenon arising in mixture model posterior inference that prevents one from meaningfully assessing posterior statistics using standard Monte Carlo procedures. This issue arises due to invariance of the posterior under actions of a group; for example, permuting the ordering of mixture components has no effect on the likelihood. We propose a resolution to label switching that leverages machinery from optimal transport. Our algorithm efficiently computes posterior statistics in the quotient space of the symmetry group. We give conditions under which there is a meaningful solution to label switching and demonstrate advantages over alternative approaches on simulated and real data.

## 1 Introduction

Mixture models are powerful tools for understanding multimodal data. In the Bayesian setting, to fit a mixture model to such data, we typically assume a prior number of components and optimize or sample from the posterior distribution over the component parameters. If prior components are exchangeable, this leads to an identifiability issue known as *label switching*. In particular, permuting the ordering of mixture components does not change the likelihood, since it produces the same model. The underlying problem is that a group acts on the parameters of the mixture model; posterior probabilities are invariant under the action of the group.

To formalize this intuition, suppose our input is a data set $X$ and a parameter $K$ denoting the number of mixture components. In the most common application, we want to fit a mixture of $K$ Gaussians to the data; our parameter set is $\Theta = \{\theta_1, \ldots, \theta_K\}$ where $\theta_k = \{\mu_k, \Sigma_k, \pi_k\}$ gives the parameters of each component. The likelihood of $x \in X$ conditioned on $\Theta$ is $p(x|\Theta) = \sum_{k=1}^{K} \pi_k f(x; \mu_k, \Sigma_k)$, where $f(x; \mu_k, \Sigma_k)$ is the density function of $\mathcal{N}(\mu_k, \Sigma_k)$. Any permutation of the labels $k = 1, \ldots, K$ yields the same likelihood. The prior is also permutation invariant. When we compute statistics of the posterior $p(\Theta|x)$, however, this permutation invariance leads to $K!$ symmetric regions in the posterior landscape. Sampling and inference algorithms behave poorly as the number of modes increases, and this problem is only exacerbated in this context since increasing the number of components in the mixture model leads to a super-exponential increase in the number of modes of the posterior. Previous methods such as the invariant losses of Celeux et al. (2000) and pivot alignments of Marin et al. (2005) do not identify modes in a principled manner.

To combat this issue, we leverage the theory of optimal transport. In particular, one way to avoid the multimodal nature of the posterior distribution is to replace each sample with its orbit under the action of the symmetry group seen as a distribution over $K!$ points. While this symmetrized distribution is invariant to group actions, we can not average several such distributions using standard Euclidean metrics. We use the notion of a Wasserstein barycenter to calculate a mean in this space, which we can project to a mean in the parameter space via the quotient map. We show conditions under which our optimization can be performed efficiently on the quotient space, thus circumventing the need to store and manipulate orbit distributions with large support.

**Contributions.** We give a practical and simple algorithm to solve the *label switching* problem. To justify our algorithm, we demonstrate that a group-invariant Wasserstein barycenter exists when the distributions being averaged are group-invariant. We give conditions under which the Wasserstein barycenter can be written as the orbit of a single point, and we explain how failure modes of our algorithm correspond to ill-posed problems. We show that the problem can be cast as computing the expected value of the quotient distribution, and we give an SGD algorithm to solve it.

## 2  Related work

**Mixture models.** Gaussian mixture models are powerful for modeling a wide range of phenomena (McLachlan et al., 2019). These models assume that a sample is drawn from one of the latent states (or components), but that the particular component assigned to any given sample is unknown. In a Bayesian setup, Markov Chain Monte Carlo can sample from the posterior distribution over the parameters of the mixture model. Hamiltonian Monte Carlo (HMC) has proven particularly successful for this task. Introduced for lattice quantum chromodynamics (Duane et al., 1987), HMC has become a popular option for statistical applications (Neal et al., 2011). Recent high-performance software offers practitioners easy access to HMC and other sampling algorithms (Carpenter et al., 2017).

**Label switching.** Label switching arises when we take a Bayesian approach to parameter estimation in mixture models (Diebolt & Robert, 1994). Jasra et al. (2005) and Papastamoulis (2015) overview the problem. Label switching can happen even when samplers do not explore all $K!$ possible modes, e.g., for Gibbs sampling. Documentation for modern sampling tools mentions that it arises in practice.[1] Label switching can also occur when using parallel HMC, since tools like `Stan` run multiple chains at once. While a single chain may only explore one mode, several chains are likely to yield different label permutations.

Jasra et al. (2005, §6) mention a few loss functions invariant to the different labelings. Most relevant is the loss proposed by Celeux et al. (2000, §5). Beyond our novel theoretical connections to optimal transport, in contrast to their method, our algorithm uses optimal rather than greedy matching to resolve elements of the symmetric group, applies to general groups and quotient manifolds, and uses stochastic gradient descent instead of simulated annealing. Somewhat ad-hoc but also related is the pivotal reordering algorithm (Marin et al., 2005), which uses a sample drawn from the distribution as a pivot point to break the symmetry; as we will see in our experiments, a poorly-chosen pivot seriously degrades the performance.

**Optimal transport.** Optimal transport (OT) has seen a surge of interest in learning, from applications in generative models (Arjovsky et al., 2017; Genevay et al., 2018), Bayesian inference (Srivastava et al., 2015), and natural language (Kusner et al., 2015; Alvarez-Melis & Jaakkola, 2018) to technical underpinnings for optimization methods (Chizat & Bach, 2018). See Solomon (2018); Peyré & Cuturi (2018) for discussion of computational OT and Santambrogio (2015); Villani (2009) for theory.

The Wasserstein distance from optimal transport (§3.1) induces a metric on the space of probability distributions from the geometry of the underlying domain. This leads to a notion of a Wasserstein barycenter of several probability distributions (Agueh & Carlier, 2011). Scalable algorithms have been proposed for barycenter computation, including methods that exploit entropic regularization (Cuturi & Doucet, 2014), use parallel computing (Staib et al., 2017), apply stochastic optimization (Claici et al., 2018), and distribute the computation across several machines (Uribe et al., 2018).

# 3 Optimal Transport under Group Actions

Before delving into technical details, we will illustrate our approach with a simple example. Assume we have some data to which we wish to fit a Gaussian mixture model with $K$ components. We can now draw samples from the posterior distribution, and we would like to obtain a point estimate of the mean of the posterior. We draw two samples $\Theta^1 = (\theta_1^1, \ldots, \theta_K^1)$ and $\Theta^2 = (\theta_1^2, \ldots, \theta_K^2)$. We cannot average them due to the ambiguity of label switching; see Figure 1(a) and §1.3 of the supplementary for a simple example. However, we can explicitly encode this multimodality as a uniform distribution over all $K!$ states:

$$\frac{1}{K!} \sum_{\sigma \in S_K} \delta_{\sigma \cdot \Theta^1} \quad \text{and} \quad \frac{1}{K!} \sum_{\sigma \in S_K} \delta_{\sigma \cdot \Theta^2}$$

where $S_K$ is the symmetry group on $K$ points that acts by permuting the elements of $\Theta^1$ and $\Theta^2$. These distributions are now invariant to permutations, so we can ask if there exists an average in this space. In this section, we prove that this is possible through the machinery of optimal transport.

We provide theoretical results relevant to optimal transport between measures supported on the quotient space under actions of some group $G$. This theory is fairly general and requires only basic assumptions about the underlying space $X$ and the action of $G$. For each theoretical result, we will use *italics* to highlight key assumptions, since they vary somewhat from proposition to proposition.

## 3.1 Preliminaries: Optimal transport

Let $(X, d)$ be a *complete* and *separable* metric space. We define the $p$-Wasserstein distance on the space $P(X)$ of probability distributions over $X$ as a minimization over matchings between $\mu$ and $\nu$:

$$W_p^p(\mu, \nu) = \inf_{\pi \in \Pi(\mu, \nu)} \int_{X \times X} d(x, y)^p \, \mathrm{d}\pi(x, y).$$

Here $\Pi(\mu, \nu)$ is the set of couplings between measures $\mu$ and $\nu$ defined as $\Pi(\mu, \nu) = \{\pi \in P(X \times X) \mid \pi(x \times X) = \mu(x), \pi(X \times y) = \nu(y)\}$.

$W_p$ induces a metric on the set $P_p(X)$ of measures with *finite* $p$-th moments (Villani, 2009). We will focus on $P_2(X)$, endowed with the metric $W_2$. This metric structure allows us to define meaningful statistics for sets of distributions. In particular, a Fréchet mean (or Wasserstein barycenter) of a set of distributions $\nu_1, \ldots, \nu_n \in P_2(X)$ is defined as a minimizer

$$\mu^* = \arg\min_{\mu \in P_2(X)} \sum_{i=1}^{n} \frac{1}{n} W_2^2(\mu, \nu_i). \tag{1}$$

We follow Kim & Pass (2017) and generalize this notion slightly, by placing a measure itself on the space $P_2(X)$. We will use $P_2(P_2(X))$ to denote the space of probability measures on $P_2(X)$ that have finite second moments and let $\Omega$ be a member of this set. Then the following functional will be finite, which generalizes (1) from finite sums to infinite sets of measures:

$$B(\mu) = \int_{P_2(X)} W_2^2(\mu, \nu) \, \mathrm{d}\Omega(\nu) = \mathbb{E}_{\nu \sim \Omega} \left[ W_2^2(\mu, \nu) \right]. \tag{2}$$

In analog to (1), a natural task is to search for a minimizer of the map $\mu \mapsto B(\mu)$. For existence of such a minimizer, we simply require that $\text{supp}(\Omega)$ is tight.

**Definition 1** (Tightness of measures). A collection $\mathcal{C}$ of measures on $X$ is called *tight* if for any $\varepsilon > 0$ there exists a compact set $K \subset X$ such that for all $\mu \in \mathcal{C}$, we have $\mu(K) > 1 - \varepsilon$.

Here are three examples of tight collections: $P_2(X)$ if $X$ is compact, the set of all Gaussian distributions with means supported on a compact space and of bounded variance, or any set of measures with a uniform bound on second moments (argued in supplementary). This assumption is fairly mild and covers many application scenarios.

Prokhorov's theorem (deferred to the supplementary) implies the existence of a barycenter:

**Theorem 1** (Existence of minimizers). $B(\mu)$ has at least one minimizer in $P_2(X)$ if $\text{supp}(\Omega)$ is tight.

## 3.2 Optimal transport with group invariances

Let $G$ be a *finite group* that acts by *isometries* on $X$. We define the set of measures invariant under group action $P_2(X)^G = \{\mu \in P_2(X) \mid g_\# \mu = \mu, \forall g \in G\}$, where the pushforward of $\mu$ by $g$ is defined as $g_\# \mu(B) = \mu(g^{-1}(B))$ for $B$ a measurable set. We are interested in the relation between the space $P_2(X)^G$ and the space of measures on the quotient space $P_2(X/G)$. If all of the measures in the support of $\Omega$ in (2) are invariant under group action, we can show that there exists a barycenter with the same property:

**Lemma 1.** *If $\Omega \in P_2(P_2(X)^G)$ is supported on the set of group-invariant measures on $X$ and $\mathrm{supp}(\Omega)$ is tight, then there exists a minimizer of $B(\mu)$ in $P_2(X)$ that is invariant under group action.*

*Proof.* Let $\mu \in P_2(X)$ denote the minimizer from Theorem 1. Define a new distribution $\mu_G = \frac{1}{|G|} \sum_{g \in G} g_\# \mu$. We verify that $\mu_G$ has the same cost as $\mu$:

$$\mathbb{E}_{\nu \sim \Omega} \left[ W_2^2 \left( \frac{1}{|G|} \sum_{g \in G} g_\# \mu, \nu \right) \right] \leq \mathbb{E}_{\nu \sim \Omega} \left[ \frac{1}{|G|} \sum_{g \in G} W_2^2(g_\# \mu, \nu) \right] \text{ by convexity of } \mu \mapsto W_2^2(\mu, \nu)$$

$$= \mathbb{E}_{\nu \sim \Omega} \left[ \frac{1}{|G|} \sum_{g \in G} W_2^2(\mu, (g^{-1})_\# \nu) \right] \text{ since } g \text{ acts by isometry}$$

$$= \frac{1}{|G|} \sum_{g \in G} \mathbb{E}_{\nu \sim \Omega} \left[ W_2^2(\mu, \nu) \right] = \mathbb{E}_{\nu \sim \Omega} \left[ W_2^2(\mu, \nu) \right] \text{ by linearity of expectation and group invariance of } \nu.$$

But $\mu$ is a minimizer, so the inequality in line 1 must be an equality. $\square$

**Remark:** If $X$ is a compact Riemannian manifold and $\Omega$ gives positive weight to the set of absolutely continuous measures, then Theorem 3.1 of Kim & Pass (2017) provides uniqueness (and this may be extended to other non-compact cases with suitable decay conditions). However, in our setting, $\Omega$ is supported on samples, measures consisting of delta functions. In this case, a simple counterexample is presented in the supplementary (§1.4) which arises in the case where $X$ consists of two points in $\mathbb{R}^2$ and $S_2$ acts to swap the points ($S_K$ is the group of permutations of a finite set of $K$ points). This is accompanied by a study of the case of $K$ points in $\mathbb{R}^d$ (see supplementary §1.3), relevant to the mixture models where components are evenly weighted and identical with a single mean parameter. Via this study we see that counterexamples seem to require a high degree of symmetry, which is unlikely to happen in applied scenarios, and does not arise empirically in our experiments.

An analogous proof technique can be used to show the following lemma needed later:

**Lemma 2.** *If $\nu_1$ and $\nu_2$ are two measures invariant under group action, then there exists an optimal transport plan $\pi \in \Pi(\nu_1, \nu_2)$ that is invariant under the group action $g \cdot \pi(x, y) = \pi(g \cdot x, g \cdot y)$.*

The above suggests that we might instead search for barycenters in the quotient space. Consider:

**Lemma 3** (Lott & Villani 2009, Lemma 5.36). *Let $p : X \to X/G$ be the quotient map. The map $p_* : P_2(X) \to P_2(X/G)$ restricts to an isometric isomorphism between the set of $P_2(X)^G$ of $G$-invariant elements in $P_2(X)$ and $P_2(X/G)$.*

We now introduce additional structure relevant to label switching. Assume that all measures $\nu \sim \Omega$ are the orbits of individual delta distributions, as they are samples of parameter values, i.e., $\nu = \frac{1}{|G|} \sum_{g \in G} \delta_{g \cdot x}$ for some $x \in X$. In the simple example of a mixture of two Gaussians from 1D data with means at $\mu_1, \mu_2 \in \mathbb{R}$, $\nu$ is of the following form $\nu = \frac{1}{2} \delta_{(\mu_1, \mu_2)} + \frac{1}{2} \delta_{(\mu_2, \mu_1)}$.

Under this assumption and by Lemmas 1 and 3, minimization of $B(\mu)$ is equivalent to finding the Wasserstein barycenter of delta distributions on $X/G$. Letting $\Omega_* := p_{*\#} \Omega$, we aim to find:

$$\underset{\mu \in P_2(X/G)}{\arg\min} \ \mathbb{E}_{\delta_x \sim \Omega_*} \left[ W_2^2(\mu, \delta_x) \right]. \tag{3}$$

From properties of Wasserstein barycenters (Carlier et al. 2015, Equation (2.9)), the support of $\mu$ lies in the set of solutions to

$$\min_{z \in X/G} \mathbb{E}_{\delta_x \sim \Omega_*} \left[ d(x, z)^2 \right] \tag{4}$$

where $d$ is the metric on the quotient space $X/G$ (see e.g. Santambrogio 2015, §5.5.5). As $\Omega$ has finite second moments, so does $\Omega_*$, giving us existence of the expectation. The existence of minimizers of $z \to \mathbb{E}_{\delta_x \sim \Omega_*}\left[d(x,z)^2\right]$ is established in §2.1 of the supplementary, giving the following lemma:

**Lemma 4.** The map $z \to \mathbb{E}_{\delta_x \sim \Omega_*}\left[d(x,z)^2\right]$ has a minimizer.

Uniqueness of minimizers is not guaranteed (see §1.4 of supplementary), but we can rewrite (3) as:

$$\underset{\mu \in P_2(X/G)}{\arg \min} \; \mathbb{E}_{\delta_x \sim \Omega_*}\left[W_2^2(\mu, \delta_x)\right] = \underset{\mu \in P_2(X/G)}{\arg \min} \int_{X/G} \int_{X/G} d(x,y)^2 \, \mathrm{d}\mu(y) \, \mathrm{d}\Omega_*(\delta_x)$$

$$= \underset{\mu \in P_2(X/G)}{\arg \min} \int_{X/G} \int_{X/G} d(x,y)^2 \, \mathrm{d}\Omega_*(\delta_x) \, \mathrm{d}\mu(y).$$

By Lemma 4, the term $y \to \int_{X/G} d(x,y)^2 \mathrm{d}\Omega_*(\delta_x)$ has a (potentially non-unique) minimizer. Call this function $b(y)$. We are left with

$$\underset{\mu \in P_2(X/G)}{\arg \min} \int_{X/G} b(y) \, \mathrm{d}\mu(y).$$

Any minimizer $y^*$ of $b$ leads to a minimizing distribution $\mu = \delta_{y^*}$, and we can conclude

**Theorem 2** (Single Orbit Barycenters). There is a barycenter solution of (2) that can be written as $\mu = \frac{1}{|G|} \sum_{g \in G} \delta_{g \cdot z^*}$.

Returning to our example of a Gaussian mixture model, we see that this theorem implies there is a barycenter (a mean in distribution space) that has the same form as the symmetrized sample distributions. Any point in the support of the barycenter is an estimate for the mean of the posterior distribution.

As an aside, we mention that our proofs do not require finite groups. In fact, we prove Theorem 2 for compact groups $G$ endowed with a Haar measure in the supplement.

**To summarize:** Label switching leads to issues when computing posterior statistics because we work in the full space $X$, when we ought to work in the quotient space $X/G$. Theorem 2 relates means in $X/G$ to barycenters of measures on $X$ which gives us a principled method for computing statistics backed by a convex problem in the space of measures: take a quotient, find a mean in $X/G$, and then pull the result back to $X$. We will see below in concrete detail that we do not need to explicitly construct and average in $X/G$, but may leverage group invariance of the transport to perform these steps in $X$.

The crux of this theory is that the Wasserstein barycenter in the setting of Lemma 1 is a point estimate for the mean of the symmetrized posterior distribution. The results leading to Theorem 2 should be understood then as a reduction of the problem of finding an estimate of the mean to that of minimizing a distance function on the quotient space; this latter minimization problem can then be solved via Riemannian gradient descent.

## 4 Algorithms

Label switching usually occurs due to symmetries of certain Bayesian models. Posteriors with the label switching make it difficult to compute meaningful summary statistics, e.g. posterior expectations for the parameters of interest.

| | |
|---|---|
| $\mathcal{M}$ | Riemannian manifold |
| $g_p$ | Inner product at $p \in \mathcal{M}$ |
| $d(p,q)$ | Geodesic distance between $p, q \in \mathcal{M}$ |
| $\mathcal{M}^K$ | $K$-fold product manifold with product metric |
| $c(p,q)$ | Transport cost, $c(p,q) = \frac{1}{2} d(p,q)^2$ |
| $\exp_p, \log_p$ | Exponential, logarithmic maps at $p \in \mathcal{M}$ |
| $S_K$ | Symmetric group on $K$ symbols |
| $C_K$ | Cyclic group on $K$ symbols |
| $\mathcal{M}/G$ | Quotient space of equivalence classes $[p] = \{g \cdot p \mid g \in G\}$ |

Table 1: Notation for our algorithm.

Any attempt to compute posterior statistics in this regime must account for the *orbits* of samples under the symmetry group. Continuing in the case of expectations, based on the previous section we can extract a meaningful notion of averaging by taking the image of each posterior sample under the symmetry group and computing a barycenter with respect to the Wasserstein metric. This resolves the ambiguity regarding which points in orbits should match, without symmetry-breaking heuristics like pivoting (Marin et al., 2005).

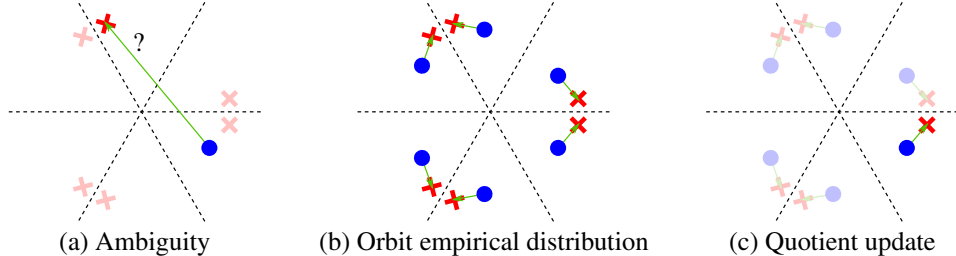

| (a) Ambiguity | (b) Orbit empirical distribution | (c) Quotient update |

Figure 1: (a) Suppose we wish to update our estimate of the average (blue) given a new sample (red) from $\Omega$; due to label switching, other points (light shade) have equal likelihood to our sample, causing ambiguity. (b) Theorem 2 suggests an unambiguous update by constructing $|G|$-point orbits as empirical distributions and doing gradient descent with respect to the Wasserstein metric. (c) This algorithm is equivalent to moving one point, with a careful choice of update functions. This schematic arises for a mean-only model with three means in $\mathbb{R}$ (§1.3 of supplementary); $G = S_3$, with action is generated by reflection over the dashed lines.

In this section, we provide an algorithm for computing the $W_2$ barycenters above, extracting a symmetry-invariant notion of expectation for distributions with label switching. As input, we are given a sampler from a distribution $\Omega$ over a space $\mathcal{M}$ subject to label switching, as well as its (finite) symmetry group $G$. Our goal is to output a barycenter of the form $\frac{1}{|G|} \sum_{g \in G} \delta_{g \cdot x}$ for some $x \in \mathcal{M}$, using stochastic gradient descent on (2). Our approach can be interpreted two ways, echoing the derivation of Theorem 2:

---

**Algorithm 1** Riemannian Barycenter of $\Omega$.

**Input:** Distribution $\Omega$, exp and log maps on $\mathcal{M}$
**Output:** Estimate of the barycenter of $\Omega$
1: Initialize the barycenter $p \sim \Omega$.
2: **for** $t = 1, \ldots$ **do**
3:     Draw $q \sim \Omega$
4:     $-D_p c(p, q) := \log_p(q)$
5:     $p \leftarrow \exp_p\left(-\frac{1}{t} D_p c(p, q)\right)$
6: **end for**

---

- The most direct interpretation, shown in Figure 1(b), is that we push forward $\Omega$ to a distribution over empirical distributions of the form $\frac{1}{|G|} \sum_{g \in G} \delta_{g \cdot x}$, where $x \sim \Omega$, and then compute the barycenter as a $|G|$-point empirical distribution whose support points move according to stochastic gradient descent, similar to the method by Claici et al. (2018).

- Since $|G|$ can grow extremely quickly, we argue that this algorithm is *equivalent* to one that moves a single representative $x$, so long as the gradient with respect to $x$ accounts for the objective function; this is illustrated in Figure 1(c).

Although our final algorithm has cosmetic similarity to pivoting and other algorithms that compute a single representative point, the details of our approach show an *equivalence* to a well-posed transport problem. Moreover, our stochastic gradient algorithm invokes a sampler from $\Omega$ in every iteration, rather than precomputing a finite sample, i.e. our algorithm deals with samples as they come in, rather than collecting multiple samples, and then trying to cluster or break the symmetry *a posteriori*.

Table 1 gives a reference for the notation used in this section. Note the Riemannian gradient of $c(p, q)$ has a particularly simple form: $-D_p c(p, q) = \log_p(q)$ (Kim & Pass, 2017).

**Gradient descent on the quotient space.** For simplicity of exposition, we introduce a few additional assumptions on our problem; our algorithm can generalize to other cases, but these assumptions are the most relevant to the experiments and applications in §5. In particular, we assume we are trying to infer a mixture model with $K$ components. The parameters of our model are tuples $(p_1, \ldots, p_K)$, where $p_i \in \mathcal{M}$ for all $i$ and some Riemannian manifold $\mathcal{M}$. We can think of the space of parameters as the product $\mathcal{M}^K$. Typically it is undesirable when two components match exactly in a mixture model, so we additionally excise any tuple $(p_1, \ldots, p_K)$ with any matching elements (together a set of measure zero). Representing parameters in a mixture model can be made through a point process, it is natural to work with the $K$th ordered configuration space of $\mathcal{M}$ considered in physics and algebraic topology (R. Fadell & Husseini, 2001):

$$\mathrm{Conf}_K(\mathcal{M}) := \mathcal{M}^K \backslash \{(p_1, \ldots, p_K) \mid p_i = p_j \text{ for some } i \neq j\} \subset \mathcal{M}^K.$$

Let $\Omega \in P(\mathrm{Conf}_K(M))$ be the Bayesian posterior distribution restricted to $\mathrm{Conf}_K(M)$ (assuming the posterior $P(\mathcal{M}^K)$ is absolutely continuous with respect to the volume measure, this restriction does essentially nothing). If $K = 1$, we can compute the expected value of $\Omega$ using a classical stochastic gradient descent (Algorithm 1). If $K > 1$, however, label switching may occur: There may be a group $G$ acting on $\{1, 2, \ldots, K\}$ that reindices the elements of the product $\mathrm{Conf}_K(M)$ without affecting likelihood. This invalidates the expectation computed by Algorithm 1.

---
**Algorithm 2** Barycenter of $\Omega$ on quotient space

**Input:** Distribution $\Omega$, exp and log maps on $\mathcal{M}$
**Output:** Barycenter $[(p_1, \ldots, p_K)]$
1: Initialize the barycenter $(p_1, \ldots, p_K) \sim \Omega$.
2: **for** $t = 1, \ldots$ **do**
3:      Draw $(q_1, \ldots, q_K) \sim \Omega$
4:      Compute $\sigma$ in (5)
5:      **for** $i = 1, \ldots, K$ **do**
6:          $-D_{p_i} c(p_i, q_{\sigma(i)}) := \log_{p_i}(q_{\sigma(i)})$
7:          $p_i \leftarrow \exp_{p_i}\left(-\frac{1}{t} D_{p_i} c(p_i, q_{\sigma(i)})\right)$
8:      **end for**
9: **end for**

---

In this case, we need to work in the quotient $\mathrm{Conf}_K(M)/G$. Two key examples for $G$ will be the symmetric group $S_K$ of permutations and the cyclic group $C_K$ of cyclic permutations. When $G = S_K$ we simply recover the $K$th unordered configuration space, typically denoted $\mathrm{UConf}_K(M)$.

$\mathrm{UConf}_K(M)$ is a Riemannian manifold with structure inherited from the product metric on $\mathrm{Conf}_K(M)$ and has the property:

$$d_{\mathrm{UConf}_K(M)}([(p_1, \ldots, p_K)], [(q_1, \ldots, q_K)]) = \min_{\sigma \in S_K} d_{\mathcal{M}^K}((p_1, \ldots, p_K), (q_{\sigma(1)}, \ldots, q_{\sigma(K)})). \quad (5)$$

The analogous fact holds for $\mathrm{Conf}_K(\mathcal{M})/G$ for other finite $G$ via standard arguments (see e.g. Kobayashi (1995)). Thus, we may step in the gradient direction on the quotient by solving a suitable optimal transport matching problem.

Since $G$ is finite, the map $\sigma$ minimizing (5) is computable algorithmically. When $G = C_K$, we simply enumerate all $K$ cyclic permutations of $(q_1, \ldots, q_K)$ and choose the one closest to $\mathbf{p}$. When $G = S_K$, we can recover $\sigma$ by solving a linear assignment problem with cost $\bar{c}_{ij} = d(p_i, q_j)^2$.

These properties suggest an adjustment of Algorithm 1 to account for $G$. Given a barycenter estimate $\mathbf{p} = (p_1, \ldots, p_K)$ and a draw $\mathbf{q} = (q_1, \ldots, q_K) \sim \Omega$: (1) align $\mathbf{p}$ and $\mathbf{q}$ by minimizing the right-hand side of (5); (2) compute component-wise Riemannian gradients from $p_i$ to $q_{\sigma(i)}$; and (3) step $\mathbf{p}$ toward $\mathbf{q}$ using the exponential map.

---
**Algorithm 3** Barycenter for Gaussian Mixtures

**Input:** Distribution $\Omega$
**Output:** Barycenter $p = (\mu_1^*, \Sigma_1^*) \ldots, (\mu_K^*, \Sigma_K^*)$

1: Initialize the barycenter $p \sim \Omega$.
2: **for** $t = 1, \ldots$ **do**
3:      Draw $((\mu_1, \Sigma_1) \ldots, (\mu_K, \Sigma_K)) \sim \Omega$
4:      Compute $\sigma$ in (5)
5:      **for** $i = 1, \ldots, K$ **do**
6:          $\mu_i^* = \mu_i^* - \eta(\mu_i^* - \mu_{\sigma(i)})$
7:          $L_i^* = L_i^* - \eta(I - T^{\Sigma_i^* \Sigma_{\sigma*(i)}}) L_i^*$
8:      **end for**
9: **end for**

---

Algorithm 2 summarizes our approach. It can be understood as stochastic gradient descent for $z$ in (4), working in space $\mathrm{Conf}_K(M)$ rather than the quotient $\mathrm{Conf}_K(M)/G$. Theorem 2, however, gives an alternative interpretation. Construct a $|G|$-point empirical distribution $\mu = \frac{1}{|G|} \sum_{\sigma \in G} \delta_{\sigma \cdot \mathbf{p}}$ from the iterate $\mathbf{p}$. After drawing $\mathbf{q} \sim \Omega$, we do the same to obtain $\nu \in P_2(\mathrm{Conf}_K(M))$. Then, our update can be understood as a stochastic Wasserstein gradient descent step of $\mu$ toward $\nu$ for problem (2). While this equivalent formulation would require $O(|G|)$ rather than $O(1)$ memory, it imparts the theoretical perspective in §3, in particular a connection to the (convex) problem of Wasserstein barycenter computation.

In the supplementary, we prove the following theorem:

**Theorem 3** (Ordering Recovery). If $\mathcal{M} = \mathbb{R}$, with the standard metric, then:

$$\mathrm{UConf}_K(M) \cong \{(u_1, \ldots, u_K) \in \mathrm{Conf}_K(\mathbb{R}) \mid u_1 < \ldots < u_K\} \subset \mathbb{R}^K.$$

Additionally, the single-orbit barycenter of Theorem 2 is unique and our algorithm provably converges.

This setting occurs when one's mixture model consists of evenly weighted components with only a single mean parameter for each in $\mathbb{R}$. The result relates our method to the classical approach of ordering these means for correspondence and shows that it is well-justified. The convergence of our algorithm leverages the convexity of $\mathrm{UConf}_K(M)$. The supplementary contains additional

discussion (§2.3) about such "mean-only" models in $\mathbb{R}^d$ for $d > 1$. They lack the niceness of the $d = 1$ case, due to positive curvature. This curvature is problematic for convergence arguments (as it leads to potential non-uniqueness of barycenters), but we empirically find that our algorithm converges to reasonable results.

**Mixtures of Gaussians.** One particularly useful example involves estimating the parameters of a Gaussian mixture over $\mathbb{R}^d$. For simplicity, assume that all the mixture weights are equal. The manifold $\mathcal{M}$ is the set of all $(\mu, \Sigma)$ pairs: $\mathcal{M} \cong \mathbb{R}^d \times \mathcal{P}^d$ with $\mathcal{P}^d$ the set of positive definite symmetric matrices. This space can be endowed with the $W_2$ metric:

$$d((\mu_1, \Sigma_1), (\mu_2, \Sigma_2))^2 = W_2^2(\mathcal{N}(\mu_1, \Sigma_1), \mathcal{N}(\mu_2, \Sigma_2)) = \|\mu_1 - \mu_2\|_2^2 + \mathfrak{B}^2(\Sigma_1, \Sigma_2), \quad (6)$$

where $\mathfrak{B}^2$ is the squared Bures metric $\mathfrak{B}^2(\Sigma_1, \Sigma_2) = \mathrm{Tr}[\Sigma_1 + \Sigma_2 - 2(\Sigma_1^{\frac{1}{2}} \Sigma_2 \Sigma_1^{\frac{1}{2}})^{\frac{1}{2}}]$.

As the mean components inherit the structure of Euclidean space, we only need to compute Riemannian gradients and exponential maps for the Bures metric. Muzellec & Cuturi (2018) leverage the Cholesky decomposition to parameterize $\Sigma_i = L_i L_i^\mathsf{T}$. The gradient of the Bures metric then becomes:

$$\nabla_{L_1} \frac{1}{2} \mathfrak{B}(\Sigma_1, \Sigma_2) = (I - T^{\Sigma_1 \Sigma_2}) L_1 \quad \text{with} \quad T^{\Sigma_1 \Sigma_2} = \Sigma_1^{-\frac{1}{2}} (\Sigma_1^{\frac{1}{2}} \Sigma_2 \Sigma_1^{\frac{1}{2}})^{\frac{1}{2}} \Sigma_1^{-\frac{1}{2}}$$

The 2-Wasserstein exponential map for SPD matrices is $\exp_\Sigma(\xi) = (I + \mathcal{L}_\Sigma(\xi))\Sigma(I + \mathcal{L}_\Sigma(\xi))$ where $\mathcal{L}_\Sigma(\xi)$ is the solution of this Lyapunov equation : $\mathcal{L}_\Sigma(\xi)\Sigma + \Sigma\mathcal{L}_\Sigma(\xi) = \xi$.

## 5 Results

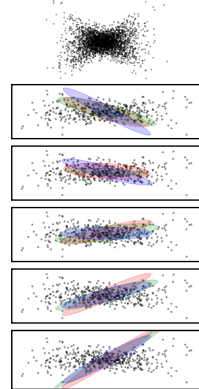

In §4, we gave a symmetry-invariant, simple, and efficient algorithm for computing a Wasserstein barycenter to summarize a distribution subject to label switching. To verify empirically that our algorithm can efficiently address label switching, we test on two natural examples: estimating the parameters of a Gaussian mixture model and a Bayesian instance of multi-reference alignment.

**Estimating components of a Gaussian mixture.** Our first scenario is estimating the parameters of a Gaussian mixture with $K > 1$ components. We use Hamiltonian Monte Carlo (HMC) to sample from the posterior distribution of a Gaussian mixture model. Naïve averaging does not yield a meaningful barycenter estimate, since the samples are not guaranteed to have the same label ordering.

Figure 2: True covariances in blue, covariances from SGD in green and pivot in red

To resolve this ambiguity, we apply our method and two baselines: the pivotal reordering method (Marin et al., 2005) and Stephens' method (Stephens, 2000). The Stephens and Pivot methods relabel samples. Stephens minimizes the Kullback–Leibler divergence between average classification distribution and classification distribution of each MCMC sample. Pivot aligns every sample to a prespecified sample (i.e. pivot) by solving a series of linear sum assignment problems. Pivot method requires pre-selecting a single sample for alignment — poor choice of the pivot sample leads to bad estimation quality, while making a "good" pivot choice may be highly non-trivial in practice. The default pivot choice is the MAP. Stephens method is more accurate, however it is expensive computationally and has large memory requirement.

To illustrate why pivoting fails, consider samples drawn from a mixture of five Gaussians with mean 0 and covariances $R_\theta M$ with $M = \left(\begin{smallmatrix} 1 & 0 \\ 0 & 0.1 \end{smallmatrix}\right)$ and $R_\theta$ a rotation of angle $\theta \in \{-\pi/12, -\pi/24, 0, \pi/12, \pi/24\}$ (Figure 2). The resulting pivot is uninformative for certain components.

|  | Pivot | Stephens | SGD |
|---|---|---|---|
| **Error (abs)** | 1.65 | 1.26 | 1.47 |
| **Time (s)** | 1.4 | 54 | 7.5 |

Table 2: Absolute error & timings

The underlying issue is that the pivot is chosen to maximize the posterior distribution. If this sample lies on the boundary of $\mathrm{Conf}_K(M)/S_K$, the pivot cannot be effectively used to realign samples. Quantitative results for this test case are in Table 2.

To get a better handle of the performance/accuracy trade-off for the three methods, we run an additional experiment. We draw samples from a mixture of five Gaussians over $\mathbb{R}^5$ with means

$0.5e_i$, where $e_i \in \mathbb{R}^5$ is the $i$-th standard basis vector with $i \in \{1, \ldots, 5\}$, and covariances $0.4I_{5 \times 5}$. We implement HMC sampler using `Stan` (Carpenter et al., 2017), with four chains discarding 500 burn-in samples and keeping 500 per chain. Then we compare the three methods, increasing the number of samples to which they have access. We measure relative error as a function of wall clock time and number of samples (Figure 3). The resulting plots align with our intuition: pivoting obtains a suboptimal solution quickly, but if a more accurate solution is desired, it is better to run our SGD algorithm.

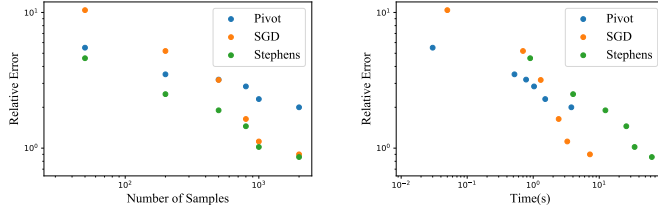

Figure 3: Relative error as a function of (a) number of samples and (b) time.

**Multi-reference alignment.** A different problem to which we can apply our methods is *multi-reference alignment* (Zwart et al., 2003; Bandeira et al., 2014). We wish to reconstruct a template signal $x \in \mathbb{R}^K$ given noisy and cyclically shifted samples $y \sim g \cdot x + \mathcal{N}(0, \sigma^2 I)$, where $g \in C_K$ acts by cyclic permutation. These observations correspond to a mixture model with $K$ components $\mathcal{N}(g \cdot x, \sigma^2 I)$ for $g \in C_K$ (Perry et al., 2017). We simulated draws from this distribution using Markov Chain Monte Carlo (MCMC), where each draw applies a random cyclic permutation and adds Gaussian noise (Figure 4a). The sampler we used was a Gibbs Sampler (Casella & George, 1992). To reconstruct the signal, we first compute a barycenter using Algorithm 2, giving a reference point to which we can align the noisy signals; we then average the aligned samples. Reconstructed signals for different $\sigma$'s are in Figure 4b. To evaluate quantitatively, we compute the relative error of the reconstruction as a function of signal-to-noise ratio $\text{SNR} = \|x\|^2 / K \sigma^2$ (Figure 4c).

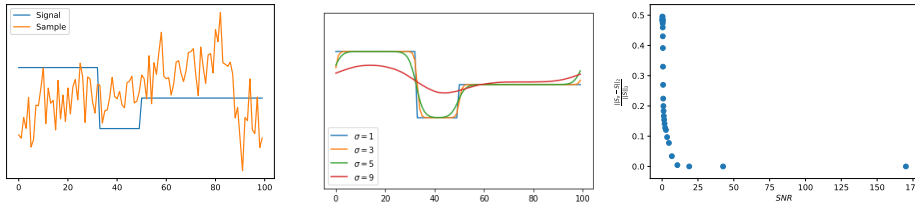

Figure 4: Reconstruction of a signal from shifted and noisy observations. (a) The true signal is plotted in blue against a random shifted and noisy draw from the MCMC chain. (b) Reconstructed signals at varying values of noise. (c) Relative error as a function of SNR.

# 6 Discussion and Conclusion

The issue underlying label switching is the existence of a group acting on the space of parameters. This group-theoretic abstraction allows us to relate a widely-recognized problem in Bayesian inference to Wasserstein barycenters from optimal transport. Beyond theoretical interest, this connection suggests a well-posed and easily-solved optimization method for alleviating label switching in practice.

The new structure we have revealed in the label switching problem opens several avenues for further inquiry. Most importantly, (4) yields a simple algorithm, but this algorithm is only well-understood when the Fréchet mean is unique. This leads to two questions: When can we prove uniqueness of the mean? More generally, are there efficient algorithms for computing barycenters in $P_2(X)^G$?

Finding faster algorithms for computing barycenters under the constraints of Lemma 1 provides an unexplored and highly-structured instance of the barycenter problem. Current approaches, such as those by Cuturi & Doucet (2014) and Claici et al. (2018) are too slow and not tailored to the demands of our application, since each measure is supported on $K!$ points and the barycenter may not share support with the input measures. Moreover, after incorporating an HMC sampler or similar piece of machinery, our task likely requires taking the barycenter of an infinitely large set of distributions. The key to this problem is to exploit the symmetry of the support of the input measures and the barycenter.

**Acknowledgements.** J. Solomon acknowledges the generous support of Army Research Office grant W911NF1710068, Air Force Office of Scientific Research award FA9550-19-1-031, of National Science Foundation grant IIS-1838071, from an Amazon Research Award, from the MIT-IBM Watson AI Laboratory, from the Toyota-CSAIL Joint Research Center, from the QCRI–CSAIL Computer Science Research Program, and from a gift from Adobe Systems. Any opinions, findings, and conclusions or recommendations expressed in this material are those of the authors and do not necessarily reflect the views of these organizations.

## Footnotes

[1] https://mc-stan.org/users/documentation/case-studies/identifying_mixture_models.html

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
