[Supplementary Material]

# Supplementary Material:
# Alleviating Label Switching with Optimal Transport

**Pierre Monteiller**
ENS Ulm
pierre.monteiller@ens.fr

**Sebastian Claici**
MIT CSAIL & MIT-IBM Watson AI Lab
sclaici@mit.edu

**Edward Chien**
MIT CSAIL & MIT-IBM Watson AI Lab
edchien@mit.edu

**Farzaneh Mirzazadeh**
IBM Research & MIT-IBM Watson AI Lab
farzaneh@ibm.com

**Justin Solomon**
MIT CSAIL & MIT-IBM Watson AI Lab
jsolomon@mit.edu

**Mikhail Yurochkin**
IBM Research & MIT-IBM Watson AI Lab
mikhail.yurochkin@ibm.com

## 1 Optimal Transport

### 1.1 Proof of Theorem 1

We first recall the definition of sequential compactness and Prokhorov's theorem, which relates it to tightness of measures:

**Definition 1** (Sequential compactness). A space $X$ is called *sequentially compact* if every sequence of points $x_n$ has a convergent subsequence converging to a point in $X$.

**Theorem 1** (Prokhorov's theorem). A collection $C \subset P_2(X)$ of probability measures is tight if and only if $C$ is sequentially compact in $P_2(X)$, equipped with the topology of weak convergence.

Now, note that the barycenter objective is bounded below by 0 and is finite, so we may pick out a minimizing sequence $\mu_n$ of $B(\mu)$. Prokhorov's theorem allows us to extract a subsequence $\mu_{n_k}$ that converges to a minimizer $\mu \in P_2(X)$ and the theorem is proved.

### 1.2 Tightness from Uniform Second Moment Bound

We argue here for a sufficient condition for tightness claimed in the text:

**Lemma 1.** If a collection of measures $\mathcal{C} \subset P_2(X)$ has a uniform second moment bound (about any reference point $x_0 \in X$), i.e.,

$$\int_X d^2(x_0, x) \, d\nu(x) < M$$

for some $M > 0$ and all $\nu \in \mathcal{C}$, then $\mathcal{C}$ is tight.

*Proof.* For any $\nu \in \mathcal{C}$ we have the following inequalities:

$$\nu\{x \mid d(x, x_0) > R\} = \int_{d(x,x_0)>R} d\nu \leq \frac{1}{R^2} \int_{d(x,x_0)>R} d(x, x_0)^2 d\nu(x) \leq \frac{M}{R^2}.$$

The last term converges to 0 as $R \to \infty$, and the set $\{x \mid d(x, x_0) \leq R\}$ is compact, so tightness follows. $\qquad\square$

Figure 1: A schematic illustrating the nontrivial part of the action of $S_3$ on $\mathbb{R}^3$. It acts on $F_3^\perp$ and the embedded 2-simplex shown via reflection over the dashed lines. One can see that reflection over these lines correspond to swapping of pairs of means, generating $S_3$ as a group.

## 1.3 Mean-only Mixture Models

Here we note some facts about mixture models, where the $K$ components are evenly weighted and identical with only one parameter each in $\mathbb{R}^d$. An example would be the simple case of a Gaussian mixture model with fixed equal covariance across each component, and a remaining unspecified mean parameter $p_i \in \mathbb{R}^d$.

In this instance, we are taking the quotient of $(\mathbb{R}^d)^K$ by an action of $S_K$ which simply permutes the $K$ factors of the product. Let us begin by investigating the case where $d = 1$. In this instance, we note that the sum of the scalar means $\sum_i p_i$ remains fixed under the action of the group. In fact, the action of the group splits into a trivial action on the 1-dimensional fixed subspace $F_K :=$ $\{(p_1, \ldots, p_k) \mid p_i \text{ all equal}\}$, and an action on $F_K^\perp$ which permutes the vertices of an embedded regular $(K-1)$-simplex about the origin. Namely, one may take the simplex in $F_K^\perp$ with vertices that consist of the point $(K-1, -1, -1, \ldots, -1)$ and its orbit. Figure 1 illustrates the concrete example of three means: $\mathbb{R}^3/S_3$. It shows $F_3^\perp$, an embedded 2-simplex, and the action of $S_3$ on this space and simplex. Section 2.2 proves that the quotient space $\mathbb{R}^K/S_K$ is a convex, easily described set, and discusses the consequences for label switching.

The splitting mentioned above is the decomposition into irreducible components. For $d > 1$, the action of $S_K$ is diagonal and acts on the $d$ components of the means $p_i$ in parallel. It preserves the scalar sum of these components over each dimension and we obtain the following splitting for the general case:

$$(\mathbb{R}^d)^K = \bigoplus_{j=1}^d \left( F_K \oplus F_K^\perp \right) \cong \mathbb{R}^d \oplus \left( \mathbb{R}^{K-1} \right)^d. \tag{1}$$

The action on the first $\mathbb{R}^d$ component is trivial, while the second component has the diagonal action permuting the vertices of an embedded regular $(K-1)$-simplex for each $\mathbb{R}^{K-1}$. The simple example of two means in $\mathbb{R}^2$ ($d = K = 2$) is discussed and illustrated in the next section (1.4), and also serves to provide a counterexample to barycenter uniqueness. For $d > 1$, the quotient $(\mathbb{R}^d)^K/S_K$ lacks the simple convexity of the $d = 1$ case, as described in Section 2.3.

## 1.4 Counterexample to uniqueness

Take $d = K = 2$ from the scenario above, which might correspond to our mixture model consisting of two Gaussians in $\mathbb{R}^2$ with equal weights and fixed variance. Only the means $(x, y; z, w) \in (\mathbb{R}^2)^2$ are taken as parameters, and the action of $S_2$ swaps the means: $(x, y; z, w) \mapsto (z, w; x, y)$. This action splits into a trivial action on $\text{Span}\{(1, 0; 1, 0), (0, 1; 0, 1)\}$ and an antipodal action ($v \mapsto -v$) on $\text{Span}\{(1, 0; -1, 0), (0, 1; 0, -1)\}$, where these are the first and second components in Eq. (1). Recall that the 1-simplex is just an interval and the action of $S_2$ merely flips the endpoints, so the antipodal action arises as the diagonal action of this flip.

The inset figure illustrates a simple schematic counterexample in the second span. The two distributions to be averaged are evenly supported on the black and white dots, invariant under reflection through the center origin $O$. Two candidate barycenters are those evenly supported on the red and

blue diamonds, and in fact, any convex combination of these two are a barycenter. This corresponds to averaging a mixture with means at $(1, 0)$ and $(-1, 0)$ and another with means at $(0, 1)$ and $(0, -1)$. Two sensible averages are a pair of means at $(0.5, 0.5)$ and $(-0.5, -0.5)$, or a pair of means at $(0.5, -0.5)$ and $(-0.5, 0.5)$.

Note that the previous example requires a high degree of symmetry for the input distributions, and uniqueness is recovered if either of the distributions are absolutely continuous. Section 2.3 further characterizes the geometry of the quotient space for $d = K = 2$, and how it leads to non-unique barycenters.

# 2 Optimal Transport with Group Invariances

## 2.1 Proof of Lemma 4

Consider an arbitrary point $z_0 \in X/G$, and we will show that a minimizer of $z \to \mathbb{E}_{\delta_x \sim \Omega_*}\left[d(x, z)^2\right]$ lies in a closed ball about $z_0$. As the function is continuous and this is a compact set, existence of a minimizer results.

By the triangle inequality, we have $d(x, z) \geq d(x, z_0) - d(z, z_0)$. Thus, we have:

$$
\begin{aligned}
\mathbb{E}_{\delta_x \sim \Omega_*}\left[d(x, z)^2\right] &= \int_{X/G} d(x, z)^2 \, \mathrm{d}\Omega_*(\delta_x) \\
&\geq \int_{X/G} (d(x, z_0) - d(z, z_0))^2 \, \mathrm{d}\Omega_*(\delta_x) \\
&= \left(\int_{X/G} d(x, z_0)^2 \, \mathrm{d}\Omega_*(\delta_x)\right) + d(z, z_0)^2 - 2d(z, z_0) \int_{X/G} d(x, z_0) \, \mathrm{d}\Omega_*(\delta_x).
\end{aligned}
$$

The last two terms are quadratic in $d(z, z_0)$. Given an arbitrary positive constant $M > 0$, some simple algebra shows that:

$$
d(z, z_0) > \frac{c + \sqrt{c^2 + 4M}}{2} \implies d(z, z_0)^2 - cd(z, z_0) > M
$$

where $c = 2\int_{X/G} d(x, z_0) \, \mathrm{d}\Omega_*(\delta_x)$. The finiteness of this integral follows from the fact that $\Omega_*$ has finite second moment, implying finite first moment. Thus, if we set $M$ to a realized value of $\mathbb{E}_{\delta_x \sim \Omega_*}\left[d(x, z)^2\right]$, we see that a minimizer lies in the ball of radius $\frac{c + \sqrt{c^2 + 4M}}{2}$ about $z_0$. Taking $z$ outside this ball implies:

$$
\begin{aligned}
\mathbb{E}_{\delta_x \sim \Omega_*}\left[d(x, z)^2\right] &\geq \left(\int_{X/G} d(x, z_0)^2 \, \mathrm{d}\Omega_*(\delta_x)\right) + d(z, z_0)^2 - 2d(z, z_0) \int_{X/G} d(x, z_0) \, \mathrm{d}\Omega_*(\delta_x). \\
&\geq d(z, z_0)^2 - 2d(z, z_0) \int_{X/G} d(x, z_0) \, \mathrm{d}\Omega_*(\delta_x) > M.
\end{aligned}
$$

## 2.2 Proof of Theorem 3

We recall the minimization problem in (5) of the paper for a sample $\mathbf{q} = (q_1, \ldots, q_K)$ and a current barycenter estimate $\mathbf{p} = (p_1, \ldots, p_K)$ (with a squared distance objective for simplicity of expression):

$$
\min_{\sigma \in S_K} d^2_{\mathbb{R}^K}((p_1, \ldots, p_K), (q_{\sigma(1)}, \ldots, q_{\sigma(K)})) = \min_{\sigma \in S_K} \sum_{i=1}^{K} \|p_i - q_{\sigma(i)}\|^2. \tag{2}
$$

Here, we invoke the monotonicity of transport in 1D (see e.g. Santambrogio (2015), Chapter 2) to see that we should simply order $\mathbf{q}$ in the same way that $\mathbf{p}$ is. That is to say: assuming $p_1 < p_2 < \ldots < p_K$ (WLOG), then the optimal $\sigma$ is such that $q_{\sigma(1)} < q_{\sigma(2)} < \ldots < q_{\sigma(K)}$.

The above argument also shows that we have a very concrete realization:

$$
\mathrm{UConf}_K(\mathbb{R}) \cong \{(u_1, \ldots, u_K) \in \mathrm{Conf}_K(\mathbb{R}) \mid u_1 < \ldots < u_K\} \subset \mathbb{R}^K.
$$

As this is an open convex set, we have uniqueness of the single-point barycenter of Theorem 2 from the paper under mild conditions on the posterior. Namely, consider that $\Omega_* \in P_2(P_2(X))$ descends to a measure $\Omega_\downarrow \in P_2(X)$, and we will need to assume that $\Omega_\downarrow$ is absolutely continuous (as you might expect). With this, Kim & Pass (2017) give us the desired result.

Furthermore, we have guaranteed convergence of stochastic gradient descent (our algorithm) in this setting, as $\mathbb{E}[W_2^2(\cdot, \nu)]$ is 1-strongly convex and the domain is convex. The next section shows us that we may not leverage such simple structure for $d > 1$.

### 2.3 Positive Curvature of Mean-Only Models

Section 1.4 shows us that in the case of $d = K = 2$:

$$\mathrm{UConf}_2(\mathbb{R}^2) \cong \mathbb{R}^2 \times C^* \qquad \text{where} \qquad C^* = (\mathbb{R}^2 \backslash \{(0,0)\})/\{v \sim -v\}.$$

$C^*$ is isometric to an infinite metric cone (2-dimensional) with cone angle $\pi$ and cone point excised. It is this positive curvature which gives rise to the counterexample presented.

More generally, 1.3 showed us that in these mean-only models there is a diagonal action on a subspace isometric to $(\mathbb{R}^{K-1})^d$. In all of these cases, under the action of $S_K$, the solid angle measure of a sphere about the origin will be divided by $K!$ when quotiented, producing a point of positive curvature, and leading to highly symmetric counterexamples with non-uniqueness of barycenters.