[Reviews · NeurIPS 2019]

Reviewer 1



The authors tackle the label ambiguity in the Bayesian mixture model setting, by considering quotients of group actions on the labels that preserve posterior likelihoods. A numerical algorithm for computing barycenters in the given latent space is then devised, by viewing the computation of a quotient metric as an optimal transport task. The paper is well written and the exposition is clear. Notably, I enjoyed the general theoretical framework provided, which is then specialized to the finite mixture model setting. Although the experimental section is quite limited, I think the framework itself makes up for it as a contribution worthy to be published. However, I encourage the authors to provide more convincing experiments. The problem of ıGı getting large quickly is acknowledged by the authors. Perhaps a simple approach to this that the authors could try, is to randomly sample elements of G for the barycentric problem and carry out a stochastic gradient descent on this level too? Specific remarks: - Line 125: I might have missed it, but is the notation S_K defined prior to this part? - Line 131: The relation looks more like equivariance than invariance. Perhaps \pi(g x, g y) can be shown to be \pi(x,y) based on the group invariance of \nu_1 and \nu_2? - Line 140: I believe the push-forward notation of \Omega with respect to p_* should be used - Line 229: Are you referring to the map \sigma that minimizes (5) here? - Line 230: Bold face p notation is not defined until line 234. - Line 266 Exponential map: The 2-Wasserstein exponential map for spd matrices is given by Exp_K(v) = (I + v)K(I + v), and the logarithm given by T^{\Sigma_1\Sigma_2} - I. The exponential map here looks more related to the exponential map under the affine-invariant metric for spd matrices. Additionally, the gradient is written to be taken with respect ot \Sigma_1, although it should be with respect to L_1 as written in Muzellec & Cuturi (2018). If the gradient is with respect to \Sigma_1, then the expression should be (I - T^{\Sigma_1\Sigma_2}) without the L_1. -Table 2: Missing a dot.

Reviewer 2



After feedback: I thank the authors for their responses but remain unconvinced that the paper should be published. I restate my main concerns below. 1) The method consists of a post-processing method for MCMC samples approximating the posterior in a mixture model. The choice of MCMC algorithm is somewhat orthogonal to the contribution of the paper, as argued by the authors, however surely what is given as an input to the proposed method must have an impact on the output. The posterior distribution in mixture models is known to be multimodal, and not only because of label switching. There is genuine multimodality (in the terminology of Jasra, Holmes & Stephens 2005). If the MCMC algorithm employed gets stuck in a local mode, then I don't see how the proposed method will give satisfactory results. Therefore, one should start with an MCMC algorithm reasonably suited to the task of sampling from multimodal target distributions. Thus HMC seems to be a very poor choice here. Even a simple random walk MH with heavy-tailed proposals would seem a better choice. 2) As far as post-processing methods for MCMC samples in the context of mixture models go (i.e. as an alternative to Stephens' method), the method has indeed some potential, although it is hard to assess without an in-depth comparison with Stephens' method. I remain unconvinced that the output of algorithm can be called a "summary", a term that is typically reserved to numbers or small dimensional vectors that summarize high dimensional or infinite-dimensional objects, e.g. the mean and the standard deviation are summaries of a probability distribution. The proposed object being a Wasserstein barycenter, it is itself a probability distribution, if I understand correctly. If I am not missing something, none of the experiments show that distribution visually. My point here is that as a reader I was expecting to see concretely what these "posterior statistics" announced in the abstract were looking like, thus I was disappointed. I believe that the changes proposed by the authors would improve the paper thus I change my rating from 4 to 5, but remain overall negative about its acceptance in NeurIPS. Pre-feedback: === The article introduces Wasserstein barycenter on the quotient space resulting from a group of isometries. The main motivation is the label switching phenomenon in mixture models, by which the posterior distribution is invariant by permutation of label indices. The part on the definition of Wasserstein barycenters on quotient spaces seems novel and might be of independent interest. However, I have concerns about the motivation from label switching and whether the article "solves" the label switching problem as claimed on page 2. The description of label switching itself is not very clear: for instance, on page 5 "Posteriors with label switching make it difficult to compute meaningful summary statistics like expectations". If the goal is to obtain a meaningful summary, I am not sure the proposed Wasserstein barycenter is a helpful contribution. Indeed it is a probability distribution, just as the posterior distribution, but it is defined on a more abstract space. So, is it really a useful summary compared to the original posterior distribution? What can be learned from the proposed barycenter that couldn't be directly learned from the posterior distribution? The method assumes that one starts with a sampler from the posterior distribution. The authors use HMC in their experiments. But a key issue with label switching is that sampling from the posterior is hard, modes might be missed, and standard MCMC methods such as HMC are likely to struggle. State of the art samplers for these settings are specifically designed to handle multimodality, e.g. parallel tempering, Wang Landau, sequential Monte Carlo, free energy MCMC, etc. In fact there is an article specifically on the fact that Hamiltonian Monte Carlo does not help tackling multimodality: https://arxiv.org/abs/1808.03230. HMC is notoriously successful in high-dimensional, log-concave settings, which is the opposite of the small-dimensional multimodal settings considered in the article. In the "multi-reference alignment" example, the authors do not specify which MCMC algorithm was used. Overall I am not convinced that the proposed manuscript solves the label switching problem. The contribution (Wasserstein barycenter on quotient spaces, gradient descent on the quotient space, etc) might be helpful to tackle other problems.

Reviewer 3



EDIT: I thank the authors for their rebuttal. As the authors promised, I stongly encourage them to clarify all notions on a simple example throughout. Some key points of the approach, like the output of the algorithm, had been understood differently by different readers. This is a sign that clarity should be improved. # Summary of the paper Label switching is the result of the invariance of some likelihood-prior pairs to a group of transformations. It is a major hindrance in obtaining meaningful point estimates in e.g., mixture models. The authors propose a novel point estimate, which is obtained as a Wasserstein barycenter of all symmetrized MCMC samples. # Summary of the review Overall, I like the paper. First, the topic is impactful; while label-switching may seem obsolete, it is still in many aspects an open problem, and it appears in many ML models, such as neural nets, when treated in a Bayesian way. The paper is well-written, the proposed method is novel, and brings a new viewpoint on the problem, as well as new tools. My main concerns are 1. I would keep a running example using a simple mixture of Gaussians, and strive to relate every abstract notion to this example. Currently, the reader has to wait until Section 4 until he understands what the authors propose in this fundamental example. This should be clear, at a high level, after the introduction. And the more abstract definitions of Sections 2 and 3 would benefit from being related to this concrete running example. 2. There lacks a discussion of how to intepret the reulting point estimates, especially in the case of non unique barycenters. What do the point estimates look like, compared to those obtained using, e.g. Stephens's algorithm with a comparable loss? # Major comments * L29 "do not identify ... in a principled manner" This statement lets the reader think that the current paper aims at a more principled way. But I could argue that your proposed method also has arbitrary components (the choice of the Wasserstein metric, for starters), which do not follow from basic principles. * L30 to L39: I would add here a paragraph to describe the proposed approach to someone who is used to MCMC for mixture models, at least in a handwavy manner. In particular, it should be clear after the introduction that you propose to take the MCMC chain, replace each sample by a symmetrized measure with finite support, and compute the Wasserstein barycenter of these measures. Correct me if I misunderstood. Currently, I had to wait until Section 5 until I had a complete idea of what the authors propose in this simple case. * Section 3: relatedly, I would add a running example to that section. For instance, keep the mixture example throughout, and explain what the different abstract objects correspond to in that example. At first read I had a hard time understanding what, say, \Omega was in the mixture example. In the end, one has to wait until Sections 4, L167 to L174, to get a grasp of what you propose. * Incidentally, am I right to think of \Omega as the collection of MCMC samples, each of which has been symmetrized and is thus now a finitely-supported measure that is invariant under G? * Section 3: the approach seems to be mainly motivated by the fact that it is convenient to take a barycenter in the quotient space. Could I also take the opposite viewpoint, and say "1) I want to perform loss-based Bayesian inference [Robert, "The Bayesian choice", 2007], 2) my action is the choice of a symmetric finite measure, 3) my loss function is the Wasserstein metric between my choice of finite measure and the symmetrized measure corresponding to the value of the parameters to infer, 4) an approximate Bayes action is thus any Wasserstein barycenter of the symmetrized history of the chain". It seems that I would obtain the same algorithm as yours, but with a clearer Bayesian motivation, what do you think? It would remain to justify the choice of the Wasserstein measure in this application, and maybe the only advantage is that again one can efficiently work on quotient spaces. * L120 to L128: with the intepretation of my previous bullet, non-uniqueness is not really a problem anymore: any minimizer has the minimum loss, and is thus an equally good Bayes action. However, it'd be interesting to exhibit several minimizers in an experiment and discuss their individual properties: some of them may be hard to interpret? * L196: "Moreover..." I did not understand that sentence. * Section 5: Figures are way too small. I understand that keeping under 8 pages is hard, but your figures are unreadable unless zooming. Also, I'd use different markers for scatterplots such as Figure 3, to be more friendly to colorblind readers and poor owners of black-and-white printers. * Section 5: is code available? A simple jupyter notebook with an illustration on 1D mixtures would help the reader make the OT notions more concrete. * Similarly, L262, I would explicitly write your algorithm in the case of mixtures of Gaussians, giving all the details (including weights) to make manifold notions more concrete. For a NIPS submission, I would aim for an imaginary reader who knows a lot about mixtures, understands the necessity of dissambiguate invariance, but who is not used to manipulating OT or manifolds, although he regularly hears about it in different contexts and will quickly follow you if you relate the latter notions to what he/she knows. # Minor comments * L23: strictly speaking, the prior also needs to be permutation-invariant for the posterior to inherit the invariance. * L111: push-forwards have not been defined so far, maybe this is the right place. * Section 4: actually, most statisticians agree that the best way to represent parameters in a mixture model is through a point process, thus naturally working with configuration spaces. Arguably, one could even say that this is the starting point of Bayesian nonparametric mixtures using Dirichlet processes. I believe this is more closely related to your use of configuration spaces than invoking algebra of physics.

[Author Response · NeurIPS 2019]

We thank all the reviewers for their insightful comments and detailed reviews. We share their enthusiasm that our work provides "a rigorous framework for dealing with label switching in mixture models" (R1) and "brings a new viewpoint on the problem, as well as new tools" (R3).

Below we discuss reviewer comments in detail. We are confident that we can address any requested revisions in time for publication to NeurIPS 2019 and that our work will be of interest to the optimal transport, Bayesian, and statistical audiences attending the conference.

**Theoretical contributions.**   **R2** asserts that the Wasserstein barycenter is no better than the original posterior distribution as a summary statistic. This is an inaccurate assessment of our work: **In all practical scenarios, the Wasserstein barycenter is a *point estimate* for the true (non-degenerate) posterior mean (Theorem 2).** The posterior alone does not easily give this information due to the inherent *label switching* phenomenon; this is the key issue addressed by our work.

**The choice of sampler is orthogonal to the problem we tackle.** For our method to succeed, it is not necessary that the sampler visits all modes of the posterior, nor does it depend on the sampler not departing from the neighborhood of a single mode. Regardless of the coverage of modes in the posterior distribution, our approach provides a principled notion of correspondence between samples from different modes, resulting in a sensible and well-posed mean estimate on the quotient space.

We will detail the choice of MCMC sampler for the multi-reference alignment experiments. We used a Gibbs sampler and then applied our SGD algorithm in these experiments.

We thank **R3** for suggesting a Bayesian interpretation of our algorithm. While nonuniqueness of the barycenter is possible (see §1.4 and §2.2 of the supplementary), this problem never occurred for us empirically. The supplementary sections and the referenced results of Arnaudon et al. 2013 suggest that uniqueness is almost surely true; nonuniqueness occurs only under an extremely high (and unlikely) degree of symmetry in posterior samples.

**Experiments.**   We are happy to provide additional experiments in our final revision, as suggested by **R1** and **R3**, and welcome any suggestions for additional experiments. We emphasize that mixture models are widely-used, effective probabilistic models in machine learning and statistics; our goal is not to improve them but rather to alleviate a common issue in Bayesian mixture modeling, which generalizes to problems with symmetry groups other than the permutation group.

As **R2** suggested, we will clarify characterizations of the baselines in the text. Next we offer a brief summary. The Stephens and Pivot methods relabel samples. Stephens minimizes the Kullback–Leibler divergence between average classification distribution and classification distribution of each MCMC sample. Pivot aligns every sample to a pre-specified sample (i.e. pivot) by solving a series of linear sum assignment problems. Pivot method requires pre-selecting a single sample for alignment — poor choice of the pivot sample leads to bad estimation quality, while making a "good" pivot choice may be highly non-trivial in practice. The default pivot choice is the MAP, however it may fail as discussed in lines 282-287 and illustrated in Figure 2. Stephens method is more accurate, however it is expensive computationally and has large memory requirement to store a tensor of size [data size $\times$ number of MCMC samples to be aligned $\times$ number of mixture components $K$].

**Clarity.**   We agree with **R3**'s suggestion that a simple running example of a mixture of Gaussians would improve clarity, and we will include one. Code will be made available via a Jupyter notebook.

- (**R1**) Line 125 : $S_K$ is the group of permutations of a finite set of $K$ points
- (**R1**) Line 131: An invariant transport plan $\pi : X \times X \to \mathbb{R}$ is invariant to the diagonal action of $G$ on $X \times X$. The invariance relation is one of equivariance if the coupling $\pi$ specifies a map, but this is not true in general. The proof strategy is correct; we will add a complete proof to the supplementary.
- (**R1**) Line 140: We were following the notation in the reference, but will change to match with the rest of the paper.
- (**R3**) Line 196: Our algorithm deals with samples as they come in, rather than collecting multiple samples and processing them together.
- (**R1**) Line 229: $\sigma$ refers to the map minimizing eq. (5).
- (**R1**) Line 230: Thanks for pointing this out. We'll fix it.
- (**R1**) Line 266: We followed the strategy of Muzellec & Cuturi and used a parameterization by factors to allow for more efficient computation of gradient steps. We will fix these inconsistencies in the final version.
- (**R2**) Page 3, eqn (1): We will use $\mu^*$ on the left hand side.
- (**R3**) Section 3, $\Omega$: interpretation of $\Omega$ suggested by **R3** is correct. Including a running example with Gaussian mixtures will help us to make the meaning of $\Omega$ more transparent.

[Meta-Review · NeurIPS 2019]

After feedback and discussion between the reviewers, the opinions of the reviewers were mostly unchanged. R3 and R1 defended the paper and the consensus was on a weak accept despite the concerns from R2. Please take into account the numerous comments of the reviewers in the final version of the paper.